
# Multilayer-HySEA model validation for landslide generated tsunamis. Part II Granular slides

Jorge Macías*, Cipriano Escalante, Manuel J. Castro

*Departamento de Análisis Matemático, Estadística e Investigación Operativa y Matemática Aplicada, Facultad de Ciencias, Universidad de Málaga, 29080-Málaga*

**Abstract**

The final aim of the present work is to propose a NTHMP-approved numerical tool for landslide generated tsunami hazard assessment. To achieve this, the novel Multilayer-HySEA model is benchmarked using laboratory experiment data for landslide generated tsunamis. In particular, this second part of the work deals with granular slides, while the first part, in a companion paper, considers rigid slides. The experimental data used have been proposed by the US National Tsunami Hazard and Mitigation Program (NTHMP) and were established for the NTHMP Landslide Benchmark Workshop, held in January 2017 at Galveston. Three of the seven benchmark problems proposed in that workshop dealt with tsunamis generated by rigid slides and are collected in the companion paper (Macías et al., 2020). Another three benchmarks considered tsunamis generated by granular slides. They are the subject of the present study. In order to reproduce the laboratory experiments dealing with granular slides, two models need to be coupled, one for the granular slide and a second one for the water dynamics. The coupled model used consists of a new and efficient hybrid finite volume/finite difference implementation on GPU architectures of a non-hydrostatic multilayer model coupled with a Savage-Hutter model. A brief description of model equations and the numerical scheme is included. The dispersive properties of the multilayer model can be found in the companion paper. Then, results for the three NTHMP benchmark problems dealing with tsunamis generated by granular slides are presented with a description of each benchmark problem.

*Corresponding author
*Email address:* jmacias@uma.es (Jorge Macías)



*Keywords:* Multilayer-HySEA model, tsunamis, granular slides, model benchmarking, landslide-generated tsunamis, NTHMP, GPU implementation

*2010 MSC:* 35L, 65-05, 76-05, 86-08

## 1. Introduction

Following the introduction of the companion paper Macías et al. (2020a), a landslide tsunami model benchmarking and validation workshop was held, January 9-11, 2017, in Galveston, TX. This workshop, which was organized on behalf of NOAA-NWS's National Tsunami Hazard Mitigation Program (NTHMP) Mapping and Modeling Subcommittee (MMS), with the expected outcome being to develop: (i) a set of community accepted benchmark tests for validating models used for landslide tsunami generation and propagation in NTHMP inundation mapping work; (ii) workshop documentation and a web-based repository, for benchmark data, model results, and workshop documentation, results, and conclusions, and (iii) provide recommendations as a basis for developing best practice guidelines for landslide tsunami modeling in NTHMP work.

A set of seven benchmark tests was selected (Kirby et al., 2018). The selected benchmarks were taken from a subset of available laboratory data sets for solid slide experiments (three of them) and deformable slide experiments (another three), that included both submarine and subaerial slides. Finally, a benchmark based on a historic field event (Valdez, AK, 1964) closed the list of proposed benchmarks. The EDANYA group (www.uma.es/edanya) from the University of Malaga participated in the aforementioned workshop, and the numerical codes Multilayer-HySEA and Landslide-HySEA were used to produce our modeled results. We presented numerical results for six out of the seven benchmark problems proposed, including the field case. The sole benchmark we did not perform at the time (due to its particular difficulty) was BP6, for which numerical results are included here.

The present work aims at showing the numerical results obtained with the Multilayer-HySEA model in the framework of the validation effort described above for the case of granular slide generated tsunamis for the complete set of





the three benchmark problems proposed by the NTHMP. However, the ultimate goal of the present work is to provide the tsunami community with a numerical tool, tested and validated, and approved by the NTHMP, for landslide generated tsunami hazard assessment. This approval has already been achieved by the Tsunami-HySEA model for the case of earthquake generated tsunamis (Macías et al., 2017; Macías et al., 2020c,d).

Fifteen years ago, at the beginning of the century, solid block landslide modeling challenged researchers and was undertaken by a number of authors (see companion paper Macías et al. (2020a) for references) and laboratory experiments were developed for those cases and for tsunami model benchmarking. In contrast, some early models (e.g., Heinrich (1992); Harbitz et al. (1993); Rzadkiewicz et al. (1997); Fine et al. (1998)) and a number of more recent models have simulated tsunami generation by deformable slides, based either on depth-integrated two-layer model equations, or on solving more complete sets of equations in terms of featured physics (dispersive, non-hydrostatic, Navier-Stokes). Examples include solutions of 2D or 3D Navier-Stokes equations to simulate subaerial or submarine slides modeled as dense Newtonian or non-Newtonian fluids (Ataie-Ashtiani and Shobeyri, 2008; Weiss et al., 2009; Abadie et al., 2010, 2012; Horrillo et al., 2013), flows induced by sediment concentration (Ma et al., 2013), or fluid or granular flow layers penetrating or failing underneath a 3D water domain (for example, the two-layer models of Macías et al. (2015) or González-Vida et al. (2019) where a fully coupled non-hydrostatic SW/Savage-Hutter model is used or the model used in Ma et al. (2015); Kirby et al. (2016) in which the upper water layer is modeled with the non-hydrostatic $\sigma$-coordinate 3D model NHWAVE (Ma et al., 2012). For a more comprehensive review of recent modeling work, see Yavari-Ramshe and Ataie-Ashtiani (2016). A number of recent laboratory experiments have modeled tsunamis generated by subaerial landslides composed of gravel (Fritz et al. (2004), Ataie-Ashtiani and Najafi-Jilani (2008), Heller and Hager (2010), Mohammed and Fritz (2012)) or glass beads (Viroulet et al., 2014). For deforming underwater landslides and related tsunami generation, 2D experiments were performed by Rzadkiewicz et al.

(1997), who used sand, and Ataie-Ashtiani and Najafi-Jilani (2008), who used granular material. Well-controlled 2D glass bead experiments were reported and modeled by Grilli et al. (2017) using the model of Kirby et al. (2016).

The benchmark problems performed in the present work are based on the laboratory experiments of Kimmoun and Dupont (see Grilli et al. (2017)) for BP4, Viroulet et al. (2014) for BP5, and Mohammed and Fritz (2012) for BP6. The basic reference for these three benchmarks, but also the three ones related to solid slides and the Alaska field case, all of them proposed by the NTHMP, is Kirby et al. (2018). That is a key reference for readers interested in the benchmarking initiative in which the present work is based on.

## 2. The Multilayer-HySEA model for granular slides

First we consider the Landslide-HySEA model, applied in Macías et al. (2015) and González-Vida et al. (2019), which for the case of one-dimensional domains reads:

$$
\begin{cases}
\partial_t h + \partial_x (hu) = 0, \\
\partial_t (hu) + \partial_x \left( hu^2 + \frac{1}{2} g h^2 \right) - g h \partial_x (H - z_s) = n_a (u_s - u), \\
\partial_t z_s + \partial_x (z_s u_s) = 0, \\
\partial_t (z_s u_s) + \partial_x \left( z_s u_s^2 + \frac{1}{2} g (1 - r) z_s^2 \right) = g z_s \partial_x ((1 - r) H - r \eta) \\
\qquad\qquad\qquad\qquad\qquad\qquad\qquad - r n_a (u_s - u) + \tau_P,
\end{cases}
\tag{1}
$$

where $g$ is the gravity acceleration ($g = 9.81 \ m/s^2$); $H(x)$ is the non-erodible (do not evolve in time) bathymetry measured from a given reference level; $z_s(x,t)$ represents the thickness of the layer of granular material at each point $x$ at time $t$; $h(x,t)$ is the total water depth; $\eta(x,t)$ denotes the free surface (measured form the same fixed reference level used for the bathymetry, for example, the mean sea surface) and is given by $\eta = h + z_s - H$; $u(x,t)$ and $u_s(x,t)$ are the averaged horizontal velocity for the water and for the granular material, respectively; $r = \frac{\rho_1}{\rho_2}$ is the ratio of densities between the ambient fluid and the granular material. The term $n_a (u_s - u)$ parameterize the friction between the




fluid and the granular layer. Finally, here we will consider $\tau_P(x,t)$ as the friction

term given in Pouliquen and Forterre (2002) to be described more precisely in

the next section.

System (1) presents the difficulty of considering the complete coupling be-

tween sediment and water, including the corresponding coupled pressure terms.

That makes its numerical approximation more complex. Moreover, it makes

also difficult to consider its natural extension to non-hydrostatic flows.

Now, if $\partial_x \eta$ is neglected in the momentum equation of the granular material,

that is, the fluctuation of pressure due to the variations of the free-surface are

neglected in the momentum equation of the granular material, then the following

weakly-coupled system could be obtained:

$$
\text{S-W system} \quad
\begin{cases}
\partial_t h + \partial_x \left( hu \right) = 0, \\[2mm]
\partial_t \left( hu \right) + \partial_x \left( hu^2 + \dfrac{1}{2}gh^2 \right) - gh\partial_x \left( H - z_s \right) = n_a(u_s - u),
\end{cases}
\tag{2}
$$

$$
\text{S-H system} \quad
\begin{cases}
\partial_t z_s + \partial_x \left( z_s u_s \right) = 0, \\[2mm]
\partial_t \left( z_s u_s \right) + \partial_x \left( z_s u_s^2 + \dfrac{1}{2}g \left( 1 - r \right) z_s^2 \right) - g \left( 1 - r \right) z_s \partial_x H = \\[2mm]
\qquad\qquad\qquad\qquad\qquad\qquad\qquad\qquad -rn_a(u_s - u) + \tau_P,
\end{cases}
\tag{3}
$$

where the first system is the standard one-layer shallow-water system and the

second one is the one layer reduced-gravity Savage-Hutter model (Savage and

Hutter (1989)), that takes into account that the granular landslide is under-

water. Note that the previous system could be also adapted to simulate sub-

aerial/submarine landslides by a suitable treatment of the variation of the grav-

ity terms. Under this formulation, it is now straightforward to improve the

numerical model for the fluid phase by including non-hydrostatic effects.



### 3. Model Equations

The Multilayer-HySEA model implements a two-phase model intended to reproduce the interaction between the slide granular material (submarine or subaerial) and the fluid. In the present work, a multi-layer non-hydrostatic shallow-water model is considered for modeling the evolution of the ambient water (see Fernández-Nieto et al. (2018)), and for simulating the kinematics of the submarine/subaerial landslide the Savage-Hutter model (3) is used. The coupling between these two models is performed through the boundary conditions at their interface. The parameter $r$ represents the ratio of densities between the ambient fluid and the granular material. Usually

$$r = \frac{\rho_f}{\rho_b}, \quad \rho_b = (1 - \varphi)\rho_s + \varphi\rho_f, \tag{4}$$

where $\rho_s$ stands for the typical density of the granular material, $\rho_f$ is the density of the fluid ($\rho_s > \rho_f$), and $\varphi$ represents the porosity ($0 \leq \varphi < 1$). In the present work, the porosity, $\varphi$, is supposed to be constant in space and time and, therefore, the ratio $r$ is also constant. This ratio ranges from 0 to 1 (i.e. $0 < r < 1$) and, even on a uniform material is difficult to estimate as it depends on the porosity (and $\rho_f$ and $\rho_s$ are also supposed constant). Typical values for $r$ are in the interval $[0.3, 0.8]$.

*The fluid model*

The ambient fluid is modeled by a multi-layer non-hydrostatic shallow-water system (Fernández-Nieto et al., 2018) to account for dispersive water waves. The model considered, that is obtained by a process of depth-averaging of the Euler equations, can be interpreted as a semi-discretization with respect to the vertical variable. In order to take into account dispersive effects, the total pressure is decomposed into the sum of hydrostatic and non-hydrostatic components. In this process, the horizontal and vertical velocities are supposed to have constant vertical profiles. The resulting multi-layer model admits an exact energy balance, and when the number of layers increases, the linear dispersion relation

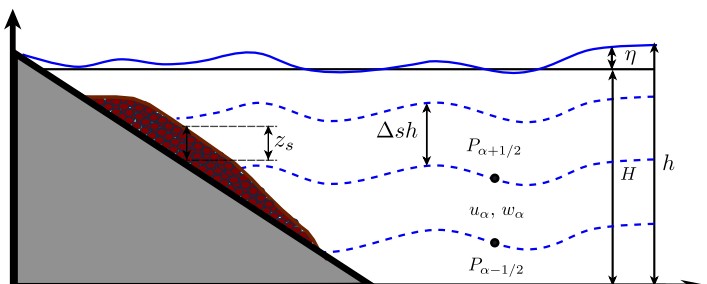

Figure 1: Schematic diagram describing the multilayer system

of the linear model converges to the same of Airy's theory. Finally, the model proposed in Fernández-Nieto et al. (2018) can be written in compact form as:

$$
\begin{cases}
\partial_t h + \partial_x (hu) = 0, \\
\partial_t (hu_\alpha) + \partial_x \left(hu_\alpha^2 + \frac{1}{2}gh^2\right) - gh\partial_x (H - z_s) \\
\qquad + u_{\alpha+1/2}\Gamma_{\alpha+1/2} - u_{\alpha-1/2}\Gamma_{\alpha-1/2} = -h\left(\partial_x p_\alpha + \sigma_\alpha \partial_z p_\alpha\right) - \tau_\alpha \\
\partial_t (hw_\alpha) + \partial_x (hu_\alpha w_\alpha) + w_{\alpha+1/2}\Gamma_{\alpha+1/2} - w_{\alpha-1/2}\Gamma_{\alpha-1/2} = -h\partial_z p_\alpha, \\
\partial_x u_{\alpha-1/2} + \sigma_{\alpha-1/2}\partial_z u_{\alpha-1/2} + \partial_z w_{\alpha-1/2} = 0,
\end{cases}
\tag{5}
$$

for $\alpha \in \{1, 2, \ldots, L\}$, with $L$ the number of layers and where the following notation has been used:

$$
f_{\alpha+1/2} = \frac{1}{2}\left(f_{\alpha+1} + f_\alpha\right), \quad \partial_z f_{\alpha+1/2} = \frac{1}{h\Delta s}\left(f_{\alpha+1} - f_\alpha\right),
$$

where $f$ denotes one of the generic variables of the system, i.e., $u$, $w$ and $p$; $\Delta s = 1/L$ and, finally,

$$
\sigma_\alpha = \partial_x \left(H - z_s - h\Delta s(\alpha - 1/2)\right), \quad \sigma_{\alpha-1/2} = \partial_x \left(H - z_s - h\Delta s(\alpha - 1)\right).
$$

Figure 1 shows a schematic picture of model configuration, where the total water height $h$ is decomposed along the vertical axis into $L \geq 1$ layers. The depth-averaged velocities in the $x$ and $z$ directions are written as $u_\alpha$ and $w_\alpha$, respectively. The non-hydrostatic pressure at the interface $z_{\alpha+1/2}$ is denoted by $p_{\alpha+1/2}$. The free surface elevation measured from a fixed reference level (for example the still-water level) is written as $\eta$ and $\eta = h - H + z_s$, where again



$H(x)$ is the unchanged non-erodible bathymetry measured from the same fixed reference level. $\tau_\alpha = 0$, for $\alpha > 1$ and $\tau_1$ is given by

$$\tau_1 = \tau_b - n_a(u_s - u_1),$$

where $\tau_b$ stands for an classical Manning-type parameterization for the bottom shear stress and, in our case, is given by

$$\tau_b = gh\frac{n^2}{h^{4/3}}u_1|u_1|,$$

and $n_a(u_s - u_1)$ accounts for the friction between the fluid and the granular layer. The latest two terms are only present at the lowest layer ($\alpha = 1$). Finally, for $\alpha = 1, \ldots, L-1$, $\Gamma_{\alpha+1/2}$ parameterizes the mass transfer across interfaces and those terms are defined by

$$\Gamma_{\alpha+1/2} = \sum_{\beta=\alpha+1}^{L} \partial_x \left(h\Delta s \left(u_\beta - \bar{u}\right)\right), \ \bar{u} = \sum_{\alpha=1}^{L} \Delta s u_\alpha$$

Here we suppose that $\Gamma_{1/2} = \Gamma_{L+1/2} = 0$, this means that there is no mass transfer through the sea-floor or the water free-surface. In order to close the system, the boundary conditions

$$p_{L+1/2} = 0, \ u_0 = 0, \ w_0 = -\partial_t \left(H - z_s\right)$$

are imposed. The last two conditions enter into the incompressibility relation for the lowest layer ($\alpha = 1$), given by

$$\partial_x u_{1/2} + \sigma_{1/2}\partial_z u_{1/2} + \partial_z w_{1/2} = 0.$$

It should be noted that both models, the hydrodynamic model described here and the morphodynamic model described in the next subsection, are coupled through the unknown $z_s$, that, in the case of the model described here, it is present in the equations and in the boundary condition ($w_0 = -\partial_t \left(H - z_s\right)$).

Some dispersive properties of the system (5) were originally studied in Fernández-Nieto et al. (2018). Moreover, for a better-detailed study on the dispersion relation (such as 'phase velocity', 'group velocity', and 'linear shoaling') the reader is referred to the companion paper Macías et al. (2020a).


Along the derivation of the two-phase model presented here, the rigid-lid
assumption for the free surface of the ambient fluid is adopted. This means
that pressure variations induced by the fluctuation on the free surface of the
ambient fluid over the landslide are neglected.

*The Landslide model*

The 1D Savage-Hutter model that it is used and implemented in the present
work is given by the system (3). The Poulliquen-Folterre friction law $\tau_P$ is given
by the expression,

$$\tau_P = -g\,(1-r)\,\mu z_s \frac{u_s^2}{|u_s|},$$

where $\mu$ is a constant friction coefficient with a key role, as it controls the
movement of the landslide. Usually $\mu$ is given by the Coulomb friction law as
the simpler parameterization that can be used in landslide models. However,
it is well-known that a constant friction coefficient does not allow to reproduce
steady uniform flows over rough beds observed in the laboratory for a range of
inclination angles. To reproduce these flows, in Pouliquen and Forterre (2002)
the authors introduced an empirical friction coefficient $\mu$ that depends on the
norm of the mean velocity $u_s$, on the thickness $z_s$ of the granular layer and on
the Froude number $Fr = \frac{u_s}{\sqrt{gz_s}}$. The friction law is given by:

$$\mu(z_s, u_s) = \begin{cases} \mu_{\text{start}}(z_s) + \left(\dfrac{Fr}{\beta}\right)^\gamma \left(\mu_{\text{stop}}(z_s) - \mu_{\text{start}}(z_s)\right), & \text{for } Fr < \beta, \\ \mu_{\text{stop}}(z_s), & \text{for } \beta \leq Fr, \end{cases}$$

with

$$\mu_{\text{start}}(z_s) = \tan(\delta_3) + (\tan(\delta_2) - \tan(\delta_1))\,exp\left(-\frac{z_s}{d_s}\right)$$

$$\mu_{\text{stop}}(z_s) = \tan(\delta_1) + (\tan(\delta_2) - \tan(\delta_1))\,exp\left(-\frac{z_s\beta}{d_sFr}\right)$$

where $d_s$ represents the mean size of grains. $\beta = 0.136$ and $\gamma = 10^{-3}$ are empir-
ical parameters. $\tan(\delta_1)$, $\tan(\delta_2)$ are the characteristic angles of the material,
and $\tan(\delta_3)$ is other friction angle related to the behavior when starting from
rest. This law has been widely used in the literature (see e.g. Brunet et al.
(2017)).



Note that the two-phase system can also be adapted to simulate subaerial

landslides. The presence of the term $(1 - r)$ in the definition of the Poulilquen-

Folterre friction law is due to the buoyancy effects, which must be taken into

account only in the case that the granular material layer is submerged in the

fluid. Otherwise, this term must be replaced by 1.

**4. Numerical Solution Method**

System (3) can be written in the following compact form:

$$\partial_t U_s + \partial_x F_s\left(U_s\right) = G_s\left(U_s\right)\partial_x H - S_s\left(U_s\right), \tag{6}$$

being

$$U_s = \begin{bmatrix} z_s \\ u_s z_s \end{bmatrix}, \; F_s\left(U_s\right) = \begin{bmatrix} z_s u_s \\ z_s u_s^2 + \dfrac{1}{2} g\left(1-r\right) z_s^2 \end{bmatrix},$$

$$G_s(U_s) = \begin{bmatrix} 0 \\ g\left(1-r\right) z_s \end{bmatrix}, \; S_s\left(U_s\right) = \begin{bmatrix} 0 \\ -r n_a(u_s - u) + \tau_P \end{bmatrix}.$$

Analogously, the multi-layer non-hydrostatic shallow-water system (5) can also

be expressed in a similar way:

$$\begin{cases} \partial_t U_f + \partial_x F_f(U_f) + B_f(U_f)\partial_x U_f = G_f(U)\partial_x(H - z_s) + \mathcal{T}_{NH} - S_f(U_f), \\ B(U_f, (U_f)_x, H, H_x, z_s, (z_s)_x) = 0, \end{cases} \tag{7}$$

where

$$U_f = \begin{bmatrix} h \\ hu_1 \\ \vdots \\ hu_L \\ hw_1 \\ \vdots \\ hw_L \end{bmatrix}, F_f(U_f) = \begin{bmatrix} h\bar{u} \\ hu_1^2 + \dfrac{1}{2}gh^2 \\ \vdots \\ hu_L^2 + \dfrac{1}{2}gh^2 \\ hu_1 w_1 \\ \vdots \\ hu_L w_L \end{bmatrix}, G_f(U_f) = \begin{bmatrix} 0 \\ gh \\ \vdots \\ gh \\ 0 \\ \vdots \\ 0 \end{bmatrix}.$$




and $B_f(U_f)\partial_x(U_f)$ contains the non-conservative products involving the momentum transfer across the interfaces and, finally, $S_f(U_f)$ represents the friction terms:

$$
B_f(U_f)\partial_x(U_f) = \begin{bmatrix} 0 \\ u_{3/2}\Gamma_{3/2} \\ u_{5/3}\Gamma_{5/2} - u_{3/2}\Gamma_{3/2} \\ \vdots \\ -u_{L-1/2}\Gamma_{L-1/2} \\ w_{3/2}\Gamma_{3/2} \\ w_{5/3}\Gamma_{5/2} - w_{3/2}\Gamma_{3/2} \\ \vdots \\ -w_{L-1/2}\Gamma_{L-1/2} \end{bmatrix}, \quad S_f(U_f) = \begin{bmatrix} 0 \\ \tau_b - n_a(u_s - u_1) \\ 0 \\ \vdots \\ 0 \end{bmatrix}.
$$

The non-hydrostatic corrections in the momentum equations are given by

$$
\mathcal{T}_{NH} = \mathcal{T}_{\mathcal{NH}}(h, h_x, H, H_x, z_s, (z_s)_x, p, p_x) = - \begin{bmatrix} 0 \\ h(\partial_x p_1 + \sigma_1 \partial_z p_1) \\ \vdots \\ h(\partial_x p_L + \sigma_L \partial_z p_L) \\ h\partial_z p_1 \\ \vdots \\ h\partial_z p_L \end{bmatrix},
$$

and finally, the operator related with the incompressibility condition at each layer is given by:

$$
B(U_f, (U_f)_x, H, H_x, z_s, (z_s)_x) = \begin{bmatrix} \partial_x u_{1/2} + \sigma_{1/2}\partial_z u_{1/2} + \partial_z w_{1/2} \\ \vdots \\ \partial_x u_{L-1/2} + \sigma_{L-1/2}\partial_z u_{L-1/2} + \partial_z w_{L-1/2} \end{bmatrix}.
$$

The discretization of systems (6) and (7) becomes difficult. In the present work, the natural extension of the numerical schemes proposed in Escalante et al. (2018b,a) is considered. These authors propose, describe and use a splitting



technique. Initially, the systems (6) and (7) are expressed as the following non-conservative hyperbolic system:

$$\begin{cases} \partial_t U_s + \partial_x F_s(U_s) = G_s(U_s)\partial_x H, \\ \partial_t U_f + \partial_x F_f(U_f) + B_f(U_f)\partial_x(U_f) = G_f(U_f)\partial_x(H - z_s). \end{cases} \quad (8)$$

Both equations are solved simultaneously using a second order HLL, positivity-preserving and well-balanced, path-conservative finite volume scheme (see Castro and Fernández-Nieto (2012)) and using the same *time step*. The synchronization of time steps is performed by taking into account the CFL condition of the complete system (8). A first order estimation of the maximum of the wave speed for system (8) is the following:

$$\lambda_{\max} = \max(|u_s| + \sqrt{(g(1 - r)z_s)}, |\bar{u}| + \sqrt{gh}).$$

Then, the non-hydrostatic pressure corrections $p_{1/2}, \cdots, p_{L-1/2}$ at the vertical interfaces are computed from

$$\begin{cases} \partial_t U_f = \mathcal{T}_{NH}(h, h_x, H, H_x, z_s, (z_s)_x, p, p_x), \\ B(U_f, (U_f)_x, H, H_x, z_s, (z_s)_x) = 0 \end{cases}$$

which requires the discretization of an elliptic operator that is done using standard second-order central finite differences. This results in a linear system than in our case it is solved using an iterative Scheduled Jacobi method (see Adsuara et al. (2016)). Finally, the computed non-hydrostatic correction are used to update the horizontal and vertical momentum equations at each layer and, at the same time, the frictions $S_s(U_s)$ and $S_f(U_f)$ are also discretized (see Escalante et al. (2018b,a)). For the discretization of the Coulomb friction term, we refer the reader to Fernández-Nieto et al. (2008).

The resulting numerical scheme is well-balanced for the water at rest stationary solution and is linearly $L^\infty$-stable under the usual CFL condition related to the hydrostatic system. It is also worth mentioning that the numerical scheme is positive preserving and can deal with emerging topographies. Finally, its extension to 2D is straightforward. For dealing with numerical experiments in


2D regions, the computational domain must be decomposed into subsets with a simple geometry, called cells or finite volumes. The 2D numerical algorithm for the hydrodynamic hyperbolic component of the coupled system is well suited to be parallelized and implemented in GPU architectures, as is shown in Castro et al. (2011). Nevertheless, a standard treatment of the elliptic part of the system do not allow the parallelization of the algorithms. The method used here and proposed in Escalante et al. (2018b,a)), makes it possible that the second step can also be implemented on GPUs, due to the compactness of the numerical stencil and the easy and massively parallelization of the Jacobi method The above-mentioned parallel GPU and multi-GPU implementation of the complete algorithm results in much shorter computational times.

## 5. Benchmark Problem Comparisons

This section presents the numerical results obtained with the Multilayer-HySEA model for the three benchmark problems dealing with granular slides and the comparison with the measured lab data for the generated water waves. In particular, BP4 deals with a 2D submarine granular slide, BP5 with a 2D subaerial slide, and BP6 with a 3D subaerial slide. The description of all these benchmarks can be found at LTMBW (2017) and Kirby et al. (2018). In the following numerical simulations, unless otherwise indicated, the quantities of the parameters are expressed in units of measure of the International System of Units. In the following of the present work all units, unless otherwise indicated, will be expressed in the International System of Units (IS).

### 5.1. Benchmark Problem 4: Two-dimensional submarine granular slide

The benchmark problem numbered as 4 reproduces the generation of tsunamis by underwater granular slides made of glass beads. The corresponding 2D laboratory were performed at the Ecole Centrale de Marseille (see Grilli et al. (2017) for a description of the experiment). A set of 58 (29 with their corresponding replicate) experiments were performed at the IRPHE (Institut de Recherche


sur les Phénomenes Hors Equilibre) precision tank. The experiments were per-

formed using a triangular submarine cavity filled with glass beads that were

released by lifting a sluice gate and then moving down a plane slope, everything

underwater. Figure 2 shows a schematic picture of the experiment set-up. The

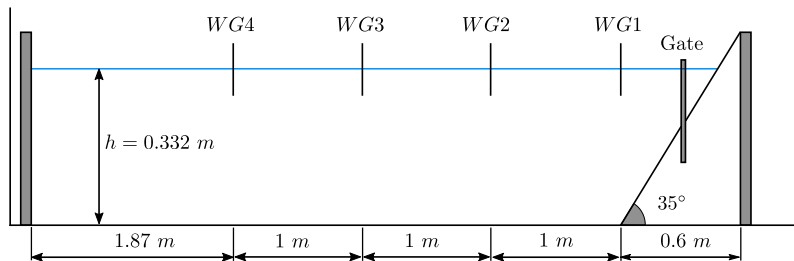

Figure 2: BP4 sketch showing the longitudinal cross section of the IRPHE's precision tank. The figure shows the location of the plane slope, the sluice gate and the 4 gages (WG1, WG2, WG3 WG4).

one-dimensional domain $[0, 6]$ is discretized with $\Delta x = 0.005\ m$ and wall bound-

ary conditions were imposed. The simulated time is $10\ s$. The $CFL$ number

was set to 0.5 and model parameters take the following values:

$$g = 9.81, \quad r = 0.78, \quad n_a = 0.2, \quad n_m = 10^{-3},$$

$$d_s = 7 \cdot 10^{-3}, \quad \delta_1 = 6°, \quad \delta_2 = 17°, \quad \delta_3 = 12°, \quad \beta = 0.136, \quad \gamma = 10^{-3}.$$

Figure 3 depicts the modeled times series for the water height at the 4 wave

gages and compared them with the lab measured data.

Figure 4 shows the location and evolution of the granular material and water

free surface at several times during the numerical simulation.

In the numerical experiments presented in this section, the number of layers

was set up to 5. Similar results were obtained with lower number of layers (4 or

3), but slightly closer to measured data when considering 5 layers. This justifies

our choice in the present test problem. Larger number of layers do not further

improve the numerical results. This may indicate that to get better numerical

results it is not longer a question related with the dispersive properties of the




model (that improve with the number of layers) but is more likely due to some missing physics.

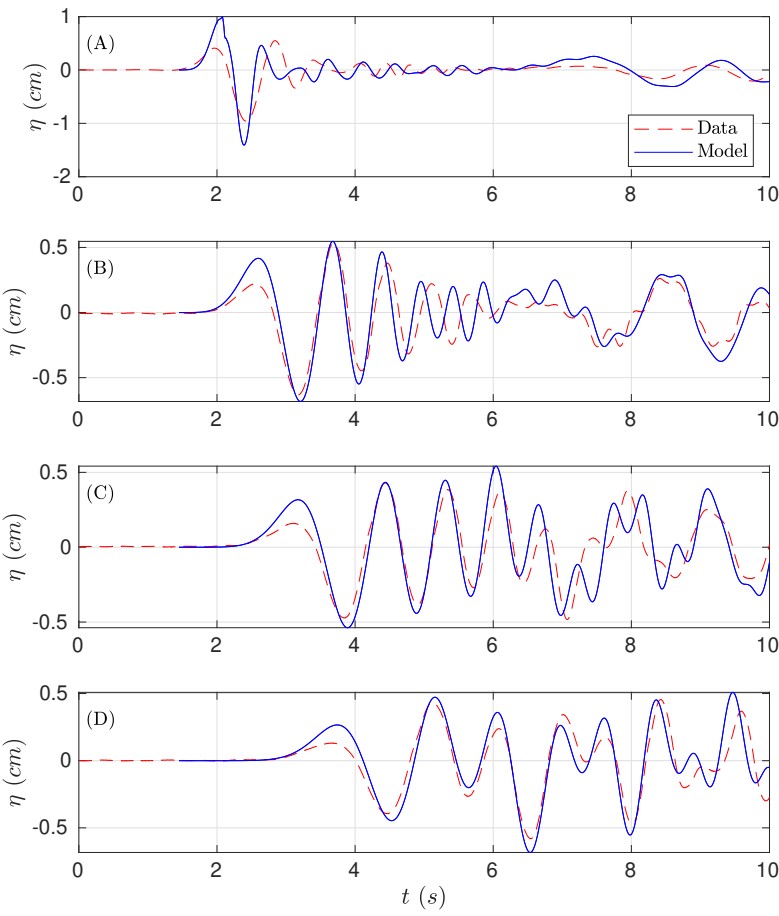

Figure 3: Comparison of numerical results (blue) with measured (red) time series at wave gauges (A) WG1, (B) WG2, (C) WG3, and (D) WG4.


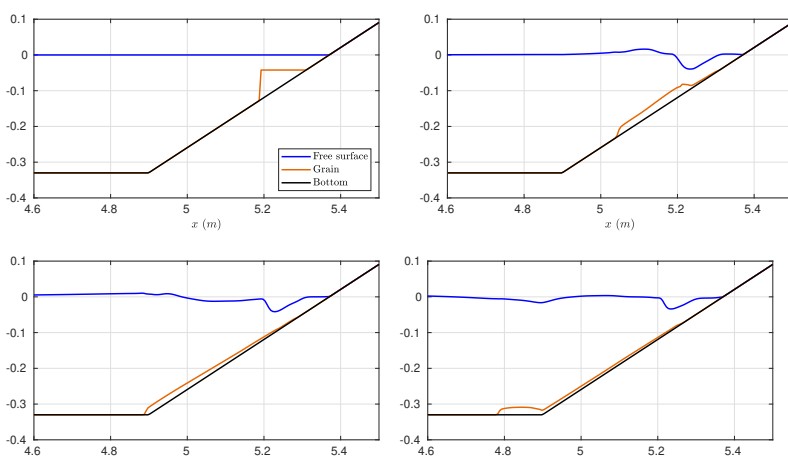

Figure 4: Modeled location of the granular material and water free surface elevation at times
$t = 0,\ 0.3,\ 0.6,\ 0.9\ s$.
*5.2. Benchmark Problem 5: Two-dimensional subaerial granular slide*

This benchmark is based on a series of 2D laboratory experiments performed

by Viroulet et al. (2014) in a small tank at the École Centrale de Marseille,

France. The simplified picture of the set-up for these experiments can be found

        in Figure 5. The granular material was confined in triangular subaerial cavities

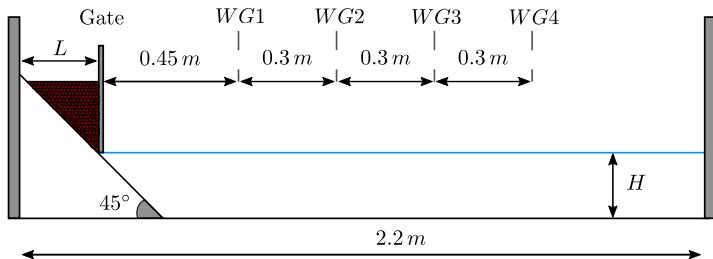

Figure 5: BP5 sketch of the set-up for the laboratory experiments.

and composed of dry glass beads of diameter $d_s$ (that was varied) and density

$\rho_s = 2,500 \, km/m^3$. This was located on a plane 45° slope just on top of the

water surface. Then the slide was released by lifting a sluice gate and entering

right away in contact with water. The experimental set-up used by Viroulet

et al. (2014) consisted in a wave tank, 2.2 m long, 0.4 m high, and 0.2 m wide.

The granular material is initially retained by a vertical gate on the dry slope.

The gate is suddenly lowered, and in the numerical experiments, it should be

assumed that the gate release velocity is large enough to neglect the time it takes

the gate to withdraw. The front face of the granular slide touches the water

surface at $t = 0$. The initial slide shape has a triangular cross-section over the

width of the tank, with down-tank length $L$, and front face height $B = L$ as the

slope angle is 45°.

For the present benchmark, two cases are considered. Case 1 defined by

the following set-up: $d_s = 1.5 \ mm$, $H = 14.8 \ cm$ and $L = 11 \ cm$ and Case 2

given by $d_s = 10 \ mm$, $H = 15 \ cm$ and $L = 13.5 \ cm$. The benchmark problem

proposed consists in simulating the free surface elevation evolution at the four

gauges WG1 to WG4 where measured data are provided, for the two test cases




described above.

The same model configuration as in the previous benchmark problem is used here. The vertical structure is reproduced using three layers in the present case. The one-dimensional domain is given by the interval $[0, 2.2]$ and it is discretized using a step $\Delta x = 0.003 \ m$. As boundary conditions, rigid walls were imposed. The simulation time is $2.5 \ s$. The $CFL$ number is set to 0.9 and model parameters take the following values:

$$g = 9.81, \quad r = 0.6, \quad n_a = 10^{-2}, \quad n_m = 9 \cdot 10^{-2},$$

$$\delta_1 = 6°, \quad \delta_2 = 26°, \quad \delta_3 = 12°, \quad \beta = 0.136, \quad \gamma = 10^{-3}.$$

Finally $d_s$ was set to $1.5 \cdot 10^{-3}$ and $10 \cdot 10^{-3}$ depending on the test case. Figure 6 shows the comparison for Case 1. In this case, the numerical results show an very good agreement when compared with lab measured data and, in particular, the two leading waves are very well captured. Figure 7 shows the comparison for Case 2. In this case, the agreement is good, but larger differences between model and lab measurements can be observed. Figure 8 shows the location of the granular material and the free surface elevation at several times for numerical simulation of Case 1. Two things can be concluded from the observation of Figures 6 and 7: (1) a much better agreement is obtained for Case 1 than for Case 2 and (2) the agreement is better for gauges located further from the slide compared with closer to the slide gauges. Although paradoxical, this second differential behavior among gauges can be explained as a consequence of the hydrodynamic component being much better resolved and simulated than the morphodynamic component (the movement of the slide material), obviously much more difficult to reproduce. But, at the same time, this implies a correct transfer of energy at the initial stages of the interaction slide/fluid.





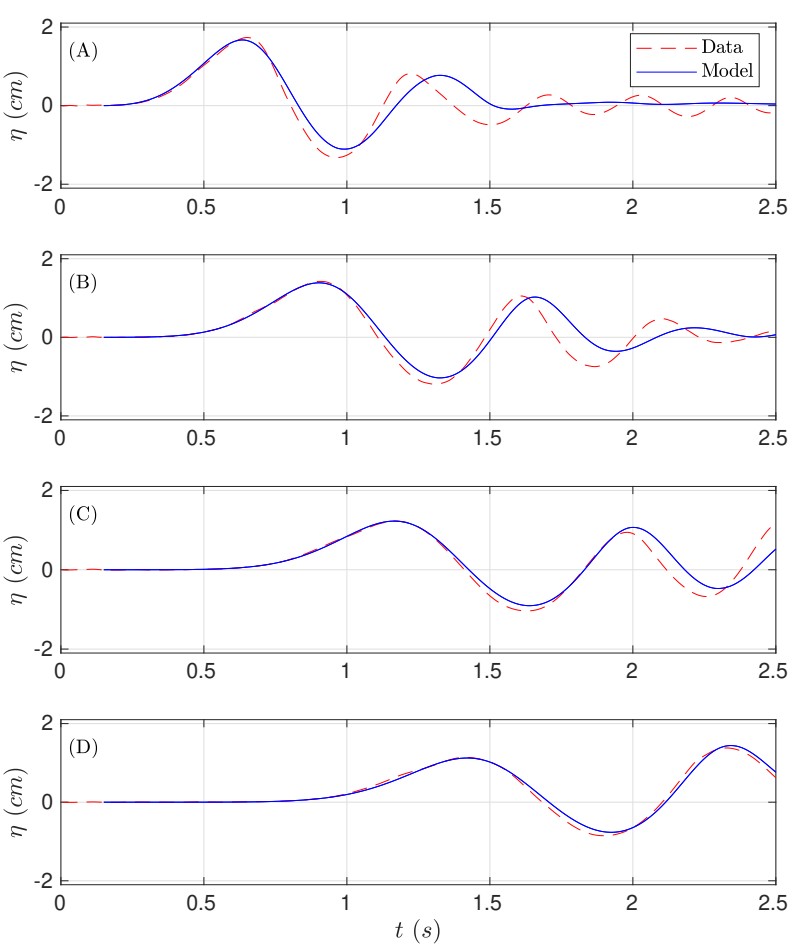

Figure 6: Numerical time series for the simulated water surface (in blue) compared with lab measure data (red). Case 1 at gauges (A) G1, (B) G2, (C) G3, and (D) G4

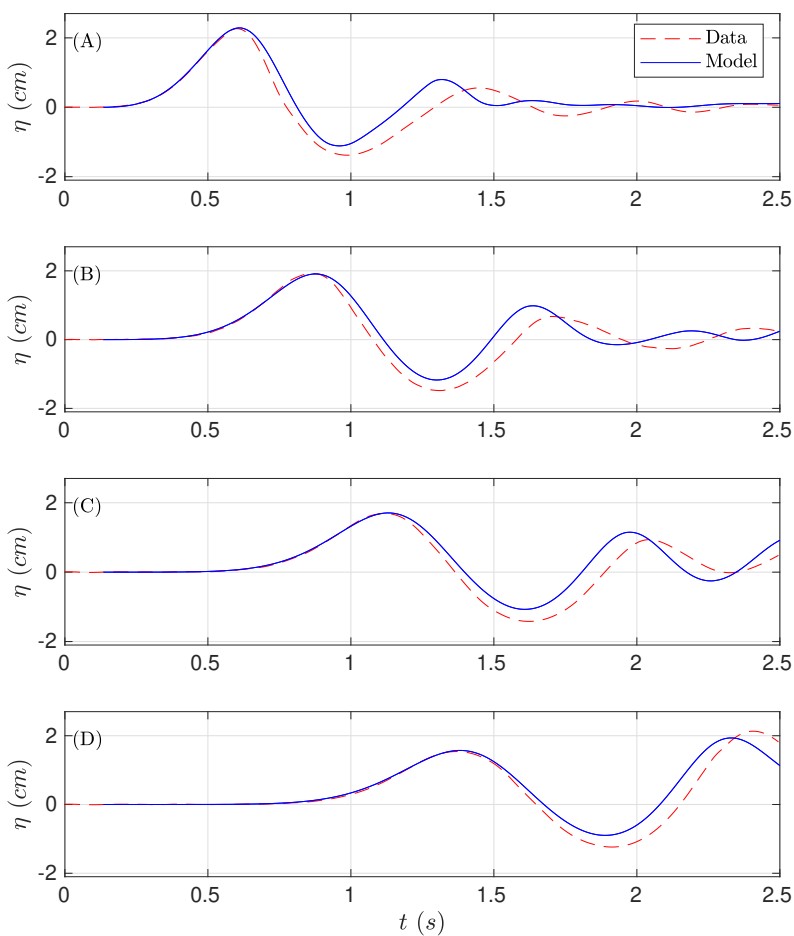

Figure 7: Numerical time series for the simulated water surface (in blue) compared with lab measure data (red). Case 2 at gauges (A) G1, (B) G2, (C) G3, and (D) G4

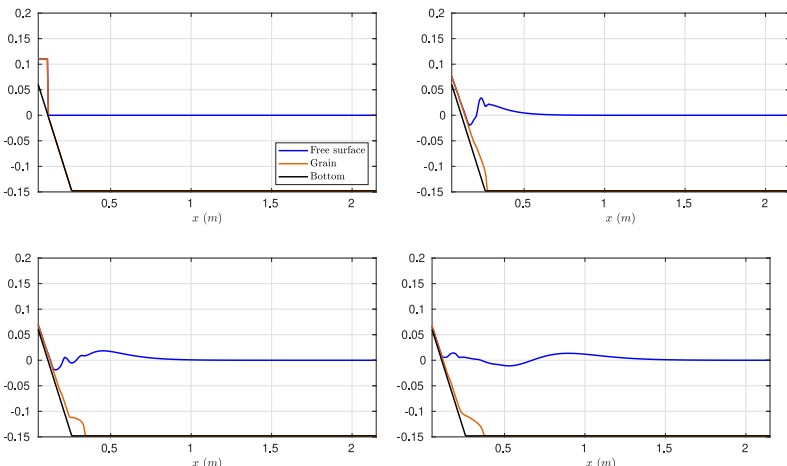

Figure 8: Modelled water free surface elevation and granular slide location at times $t =$ 0, 0.2, 0.4, 0.8 $s$ for the Case 1.

### 5.3. Benchmark Problem 6: Three-dimensional subaerial granular slide

This benchmark problem is based on the 3D laboratory experiment of Mohammed and Fritz (2012) and Mohammed (2010). Benchmark 6 simulates the rapid entry of a granular slide into a 3D water body. The landslide tsunami experiments were conducted at Oregon State University in Corvallis. The landslides are deployed off a plane 27.1° slope, as shown in Figure 9. The landslide

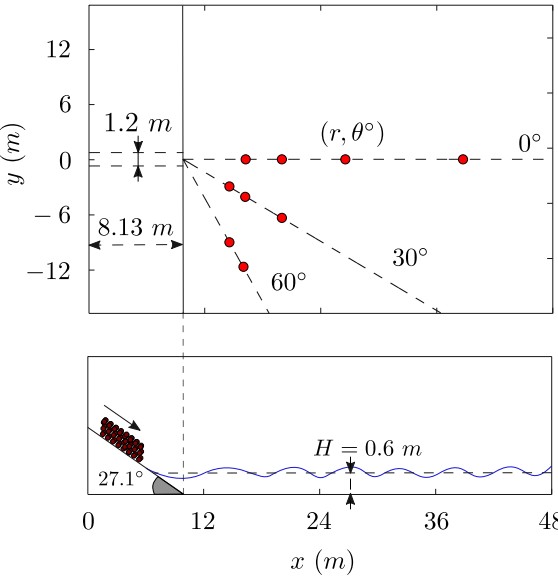

Figure 9: Schematic picture of the computational domain. Plan view in the upper pannel. Cross-section at $y = 0\ m$ in the lower pannel. The red dots represent the distribution of the wave gauge positions in the laboratory set-up.

material is deployed using a box measuring $2.1\ m \times 1.2\ m \times 0.3\ m$, with a volume of $0.756\ m^3$ and weighting approximately $1360\ kg$. The case selected by the NTHMP as benchmarking test is the one with a still water depth of $H = 0.6\ m$ (see Figure 9). The computational domain is the rectangle defined by $[0, 48] \times [-14, 14]$, and it is discretized with $\Delta x = \Delta y = 0.06\ m$. At the boundaries, wall boundary conditions were imposed. The simulation time is $20\ s$ and we set the $CFL = 0.5$. According to Mohammed and Fritz (2012) and




Mohammed (2010), the three-dimensional granular landslide parameters were set to

$$g = 9.81, \quad r = 0.55, \quad n_a = 4, \quad , n_m = 4 \cdot 10^{-2},$$

$$d_s = 13.7 \cdot 10^{-3}, \quad \delta_1 = 6°, \quad \delta_2 = 30°, \quad \delta_3 = 12°, \quad \beta = 0.136, \quad \gamma = 10^{-3}.$$

The vertical structure of the fluid layer is modeled using three layers. Similar results were obtained with 2 layers.

In the beginning, the slide box is driven using four pneumatic pistons. Here we provide comparisons for the case of pressure in the pneumatic pistons of the landslide tsunami generator of $P = 0.4$ MPa ($P = 58$ PSI). In Mohammed (2010), it is shown that for this test case, the landslide box velocity reached a velocity of $v_b = 2.3 \cdot \sqrt{g \cdot 0.6} = 5.58$ $m/s$ that serve us as a constant initial condition for the $x$-component of $u_s$ wherever $z_s > 0$.

The benchmark problem proposed consists in simulating the free surface elevation at some wave-gauges. In the present study, we include the comparison for the 9 wave gauges displayed in Figure 9 as red dots. A total number of 21 wave gauges composed the whole set of data, plus 5 run-up gauges. The wave-gauge in coordinates $(r, \theta°)$ are given more precisely in Table 1. Before

| $\theta°$ | 0° | | | | 30° | | | 60° | |
|---|---|---|---|---|---|---|---|---|---|
| $r$ | 5.12 | 8.5 | 14 | 24.1 | 3.9 | 5.12 | 8.5 | 3.9 | 5.12 |

Table 1: Location of the 9 waves gauges referenced to the toe's slope.

comparing time series, we first check the simulated landslide velocity at impact with the measured one. The slide impact velocity measured in the lab experiment is 5.72 $m/s$ at time $t = 0.44$ $s$. The numerically computed slide impact velocity is slightly underestimated with a value of 5.365 $m/s$ at time $t = 0.4$ $s$ as it can be seen in the upper panel of Figure 10. The final simulated grain deposit is located partially on the final part of the sloping floor and partially at the flat bottom closer to the point of change of slope as it is shown in the lower panel of Figure 10. This can be compared with the actual final location of the granular material in the experimental setup. The simulated deposits extend



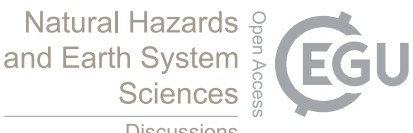

further, being thinner. This is probably due to the fact that we are neglecting

the friction that it is produced by the change in the slope at the transition area.

In Ma et al. (2015) a similar result and discussion can be found. Figure 11

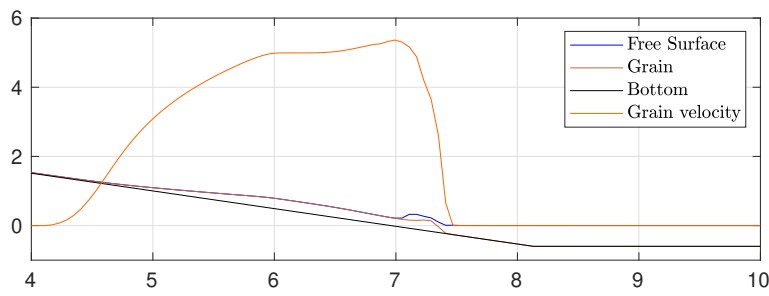

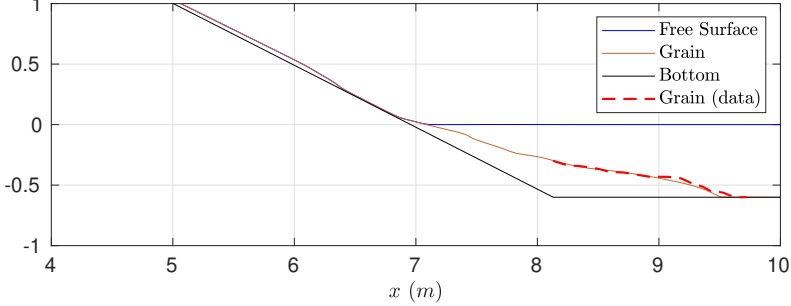

Figure 10: Cross-section at $y = 0$ $m$. at the landslide impact time $t = 0.4$ $s$ (up) and at $t = 20$ $s$ (down)

presents the comparisons between simulated and the measured waves at the 9

gauges we have retained. Model results are in good agreement with measured

time. Despite this, wave heights are overestimated at some stations, specially

those closer to the shoreline (for example, the station with $\theta = 30°$ and $r = 3.9$).

This effect has been also observed and discussed in Ma et al. (2015). At some

of the time series, it can be observed that the small free-surface oscillations at

the final part of the time series, are not well-captured by the model. This is

partially due to the relatively coarse horizontal grids used in the simulation.

These same behaviour can be also observed in Figure 12 in this case for the




comparisons between simulated and measured run-up values at some measure

locations situated at the shoreline (as for $x = 7.53$).

Table 2 shows the wall-clock times on a NVIDIA Tesla P100 GPU. In can be

observed that including non-hydrostatic terms in the SWE-SH system results in

an increase of the computational time in 2.9 times. If a richer vertical structure

is considered, then larger computational times are required. As examples for the

two and three-layer systems, 3.48 and 4.66 times increase in the computational

effort.

|          | Runtime (s) | Ratio |
|----------|-------------|-------|
| SWE-SH   | 186.55      | 1     |
| 1L NH-SH | 541.11      | 2.9   |
| 2L NH-SH | 649.19      | 3.48  |
| 3L NH-SH | 869.32      | 4.66  |

Table 2: Wall-clock times in seconds for the SWE-SH and the non-hydrostatic GPU implementations. The ratios are with respect the SWE-SH model implementation.

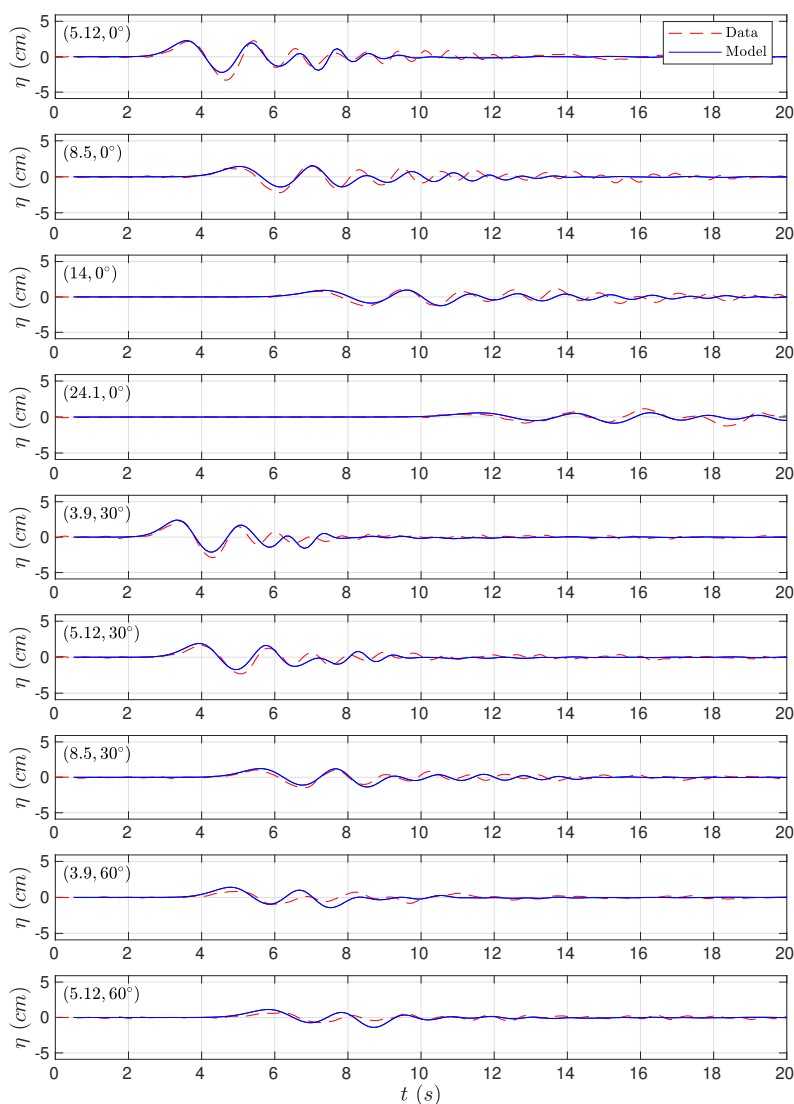

Figure 11: Simulated (solid blue lines) time series compared with measured (dashed red lines) free surface waves for the 9 wave gauges considered.
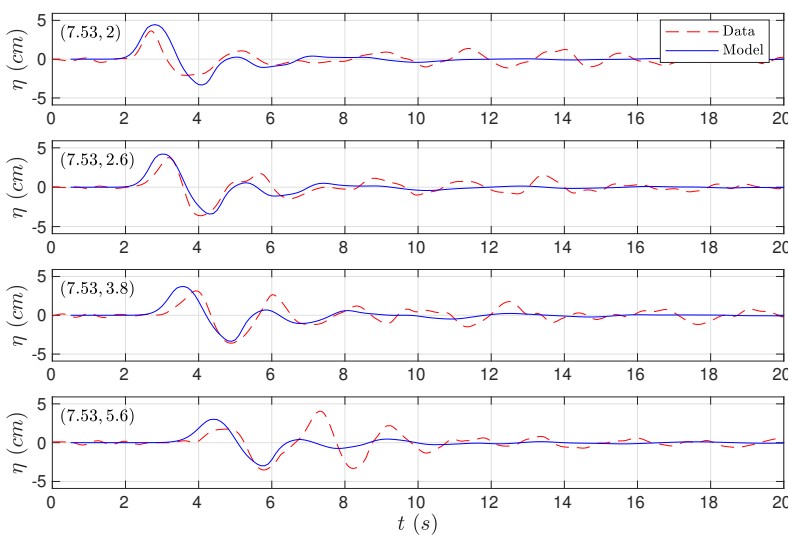

Figure 12: Time series comparing numerical run-up (solid blue) at the 4 run-up gauges with
the measured (dashed red) data.



## 6. Concluding Remarks

Numerical models need to be validated previous to their use as predictive tools. This requirement becomes even more necessary when these models are going to be used for risk assessment in natural hazards where human lives are involved. The present work aims to benchmark the novel Multilayer-HySEA model for landslide generated tsunamis produced by granular slides, in order to provide in the future to the tsunami community with a robust, efficient and reliable tool for landslide tsunami hazard assessment.

The Multilayer-HySEA code implements a two-phase model to describe the interaction between landslides (aerial or subaerial) and water body. The upper phase describes the hydrodynamic component. This is done using a stratified vertical structure that includes non-hydrostatic terms in order to include dispersive effects in the propagation of simulated waves. The motion of the landslide is taken into account by the lower phase, consisting of a Savage-Hutter model. To reproduce these flows, the friction model given in Pouliquen and Forterre (2002) is considered here. The hydrodynamic and morphodynamic models are weakly-coupled through the boundary condition at their interface.

The implemented numerical algorithm combines a finite volume path-conservative scheme for the underlying hyperbolic system and finite differences for the discretization of the non-hydrostatic terms. The numerical model is implemented to be run in GPU architectures. The two-layer non-hydrostatic code coupled with the Savage-Hutter use here, has been shown to run at very efficient computational times. To assess this, we compare with respect to the one-layer SWE/Savage-Hutter GPU code. For the numerical simulations performed here, the execution times for the non-hydrostatic model are always below 4.66 times the times for the SWE model for a number of layers up to three. We can conclude that the numerical scheme presented here is very robust, extremely efficient, and can model dispersive effects generated by submarine/subaerial landslides at a low computational cost considering that dispersive effects and a vertical multi-layer structure are included in the model. Model results show a good agreement



with the experimental data for the three benchmark problems considered. In particular, for BP5, but this also occurs for the other two benchmark problems. In general, it is shown a better agreement for the hydrodynamic component, compare with their morphodynamic counterpart, which is more challenging to reproduce.

## 7. Code and data availability

The numerical code is currently under development and only available to close collaborators. In the future, we will provide an open version of the code as we already do for Tsunami-HySEA. This version will be downloaded from https://edanya.uma.es/hysea/index.php/download.

All the data used and necessary to reproduce the set-up of the numerical experiments and the laboratory measured data to compared with, can be downloaded from LTMBW (2017) at http://www1.udel.edu/kirby/landslide/. Finally, the NetCDF files containing the numerical results obtained with the Multilayer-HySEA code for all the tests presented here can be found and download from Macías et al. (2020b).

## 8. Authors' contributions

JM is leading the HySEA codes benchmarking effort undertaken by the EDANYA group, he wrote most of the paper, reviewed and edited it, assisted in the numerical experiments and in their set up. CE implemented the numerical code and performed all the numerical experiments, he also contributed to the writing of the manuscript. JM and CE did all the figures. MC strongly contributed to the design and implementation of the numerical code.

## 9. Competing interest

The authors declare that they have no conflict of interest.





## 10. Acknowledgements

This research has been partially supported by the Spanish Government-FEDER funded project MEGAFLOW (RTI2018-096064-B-C21), the Junta de Andalucía-FEDER funded project UMA18-Federja-161 and the University of Málaga, Campus de Excelencia Internacional Andalucía Tech.

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
