# Peer review of "Multilayer-HySEA model validation for landslide generated tsunamis. Part II Granular slides"

_Natural Hazards and Earth System Sciences, 2020_

## Referee Comment (RC1) · Anonymous Referee #1 · 6 Oct 2020

General Comments:

As was the case for the first part of this paper, this is once again an informative paper that allows the modeling community to get a feeling of the reliability of the simulations performed with the Landslide-HySea and Multi-layer HySea model when used for real-life hazard assessment studies. This work presents results of modeling 3 of 7 benchmark problems proposed by the National Tsunami Hazard Mitigation Program (NTHMP). The three problems are based on data collected via experimental studies on tsunami generation by 2-D and 3-D, deformable slides from aerial and subaerial initial positions. The Landslide-HySea version of the code is used to represent the slide dynamics by means of a Savage-Hutter approach. This is coupled with the Multi-layer HySea version to capture the dispersive dynamics of the hydrodynamic phase.

The topic and results presented in the paper are within the scope NHESSD topics. The paper provides a sufficiently (perhaps too much) detailed description of both, the governing equations modeled in the code and the numerical algorithm implemented to resolve the system. Additional references are provided for readers interested in additional details. Given the complexity of the mathematical modeling and numerical scheme employed to solve the equations, one wonders whether the reader should be referred to a separate publication for that explanation, and the manuscript could be focused on the modeling setup and results alone. The authors provide and adequate literature review of pre-existing validation efforts in the introductory section of the report. They also provide a description of the numerical implementation of the laboratory experiments used for the validation. The results of all three experiments are presented in a clear and concise manner.

I have not really found any major issues with the paper and I am ready to recommend publication with very minor modifications:

Specific Modifications:

-pp1 l.7: Here and in a other location(s) in the paper the term "approved" is used to refer to the NTHMP process of testing codes. It should be corrected to "validated" or "tested" as the NTHMP does not "approve" or "certify" any models. Please check with NYHMP for clarification if needed.

-pp1 l15-17: The authors mention the workshop consisted of 7 benchmark problems (3 were presented in Part I of the paper and 3 in this Part II), it would be good to explain if the missing problem was attempted and what results were obtained.

-pp 8, l181-182: Please, specify what boundary condition is applied where for each of the three BCs specified in the equations.

-pp14,18, 23: For all benchmark problems, please specify how parameters (r, na, nm,...) were selected or whether they were provided with the data. Also explain how

Dx (delta x), Dy (delta y) is selected

-pp17 l313: Please, correct units of density (km/mˆ3)

-pp17, l316: Please, replace "consists in" with "consists of", wherever it appears in the paper.

-pp23, l370: It would be interesting to know whether the non-dispersive case of 1-layer was attempted and how the results would compare with the multi-layer cases. If available, please add.

-p23, l371-376: The description of how the slide is initiated is unclear. Please, explain with more detail. Is the entry velocity specified? If not, how it is reached? What is the function of the pneumatic pistons?

-pp24, Figure 10: I would suggest using more distinguishable colors for the lines in the top panel, it is hard to tell the Grain from the Grain Velocity lines. -Please, be more detailed in the legend, specify what magnitude is represented by "Grain". -Does the vertical axes represent position or velocity? Perhaps, the left axes should be used for distance and the right one for velocity?

-pp27, Figure 12: Please specify if number in top left corner refer to x-, y= (positions)

Some stylistic corrections (these are some of the corrections needed, but not all, please scan the document for additional typos):

-pp4, l 99: Please correct to " initiative which the present work is based on"

-pp6, l144-147: move : "…, the ratio r is also constant (rho_f and rho_s are also constant)" from line 147 to line 145.

-pp6, l154: Please replace "vertical variable" with "vertical coordinate".

-pp9, l197: Correct: "The Savage-Hutter model that is used and …."

-pp12, l238: Please, spell out "HLL"

-pp13, l276-280: The first sentence is repeated almost literally. Please, correct.

-pp14, l304: What is meant by "no longer"?, Please word correctly.

-pp25, l405:, Correct: "In can be..."

pp.28: l415: Please, correct: to "The present work aims at benchmarking the model..."

pp.28, l432: Correct to: " Savage-Hutter used here".

pp.29, l452: Correct to: "data compared with, ..."

---

## Referee Comment (RC2) · Anonymous Referee #2 · 16 Oct 2020

In this manuscript, the authors validate their new wave-slide model Multilayer-HySEA against to three granular landslide benchmarks from US National Tsunami Hazard and Mitigation Program (NTHMP). Their work focuses on presenting the deformable Savage-Hutter type landslide module additional to the wave model framework. The subject of the manuscript is important to both research of wave modeling and landslide tsunami hazard. The model has certain innovation by allowing the slide model and the overlying wave model to be run in the same Cartesian coordinate system. Their simulation results show that the model can capture landslide-generated waves well with a deformable granular slide and the simulations are quite efficient with the GPU-parallel computing technique. However, I find the paper to be somewhat lacking substance on its own. Some questions are necessary to be answered in the manuscript before it is

published:

1. Are the governing equations of deformable landslide derived in local coordinate or Cartesian coordinate? In most of landslide cases, steep slopes are involved in the bathymetry. If the landslide governing equations are derived in Cartesian coordinate, the assumption in regular shallow-water-type equations that the pressure gradient in vertical momentum equation is balanced by gravity force is not valid anymore. It needs non-hydrostatic pressure to represent the vertical acceleration of the landslide. If the landslide governing equations are derived in local coordinate and then transformed into Cartesian coordinate, the problem will not be there. The vertical acceleration perpendicular to the local bed is negligible.

2. What is the physical connection of the parameters in the landslide model to the material properties? How sensible is the change of these parameters to the landslide motion and induced tsunamis? In all three benchmarks, the internal friction angles and basal friction angles were. In the present model, another set of characteristic angles were used. The author should provide the relation between the angles used in the simulations and the real material properties. The parameter denoting the buoyancy effect r is just a tuning parameter if the author cannot link it to the status of real landslide. What is the model user to set up this parameter If the user does not know any field or experimental observations ahead of time?

Minor suggestions: 1. I am not sure if the authors' use of the term 'two-phase model' is correct or not. Most of the time only the model solving both the fluid phase and solid phase of the landslide is called 'two-phase model'. 2. In Line 54 the authors mention 'particular difficulty' of the BP6 simulations. If they do not say somewhere what the difficulty is, they can simply delete the words in the brackets. 3. Both BP5 and BP6 provide the slide shape at different time in the experiments. I would recommend the authors to compare the slide shape of simulation results to the observations in Figure 4 and Figure 8. 4. The symbol r is used twice. It both denotes the slide liquefaction and the wave gauge locations in Table 1. 5. Some typos: 2020a in line 17; duplicate

sentences from line 276 to 280.

---

## Author Comment (AC1) · 9 Dec 2020

N.B.: A compiled pdf version of the response is attached to check the references not appearing in the automatic compilation of the system.

Anonymous Referee #2

In this manuscript, the authors validate their new wave-slide model Multilayer-HySEA against to three granular landslide benchmarks from US National Tsunami Hazard and Mitigation Program (NTHMP). Their work focuses on presenting the deformable Savage-Hutter type landslide module additional to the wave model framework. The

subject of the manuscript is important to both research of wave modeling and landslide tsunami hazard. The model has certain innovation by allowing the slide model and the overlying wave model to be run in the same Cartesian coordinate system. Their simulation results show that the model can capture landslide-generated waves well with a deformable granular slide and the simulations are quite efficient with the GPU-parallel computing technique. However, I find the paper to be somewhat lacking substance on its own. Some questions are necessary to be answered in the manuscript before it is published:

1. Are the governing equations of deformable landslide derived in local coordinate or Cartesian coordinate? In most of landslide cases, steep slopes are involved in the bathymetry. If the landslide governing equations are derived in Cartesian coordinate, the assumption in regular shallow-water-type equations that the pressure gradient in vertical momentum equation is balanced by gravity force is not valid anymore. It needs non-hydrostatic pressure to represent the vertical acceleration of the landslide. If the landslide governing equations are derived in local coordinate and then transformed into Cartesian coordinate, the problem will not be there. The vertical acceleration perpendicular to the local bed is negligible.

*The governing equations of the landslide motion are derived in Cartesian coordinates. In some cases where steep slopes are involved, landslide models based on local coordinates allow representing the slide motion better. However, when general topographies are considered and not only simple geometries, landslide models based on local coordinates also introduce some difficulties on the final numerical model and on its implementation. Besides, the computational component is important.*

*In this work, we focus on the hydrodynamic part, and that is one of the reasons for choosing a simple landslide model based on Cartesian coordinates. Of course,*

*the strategies presented here can also be adapted for more sophisticated landslide models.*

*For example, in () a non-hydrostatic model for the hydrodynamic part that is similar to the one presented here for the case of a single layer was introduced. In the aforementioned paper, the authors study the influence of coupling the hydrodynamic model with a granular model that is derived in both coordinates references: Cartesian and local coordinates. The front positions calculated with the Cartesian model progress faster and, after some time, they are slightly ahead compared with the local coordinate model solution (see, for instance, Figure 4 in ()). This is due to the fact that the Cartesian model uses the horizontal velocity instead of the velocity tangent to the topography. In any case, the differences between the two models are not very noticeable.*

*A granular slide model based on local coordinates might gives better results. However, when introducing a non-hydrostatic pressure, the model is closer to a 3D solver. In such a case, the influence on the reference coordinate system barely exists. That is the reason why in (), both non-hydrostatic models based on different coordinate systems show similar results.*

*In any case, although on the present work we focus on the hydrodynamic part, we can appreciate on the benchmark tests that the numerical results show a very good agreement with the lab experiments, despite the simple landslide model chosen here.*

*At the end of section 2, a paragraph has been added including reviewer's comment and our response to it.*

2. What is the physical connection of the parameters in the landslide model to the material properties? How sensible is the change of these parameters to the landslide motion and induced tsunamis? In all three benchmarks, the internal

friction angles and basal friction angles were. In the present model, another set of characteristic angles were used. The author should provide the relation between the angles used in the simulations and the real material properties.

*The parameters involved at each simulation are:*

$$g, \ r, \ n_a, \ n_m, \ d_s, \ \delta_i, \ \beta, \text{ and } \gamma.$$

*The parameters $g, \ r, \ n_m,$ and $d_s$ are related to physical settings given at each experiment. $\beta$ and $\gamma$ are empirical parameters that were chosen as in the seminal paper ().*

*The friction angle $\delta_1, \ \delta_2$ are characteristic angles of the material, and $\delta_3$ is related to the behavior of the slide motion when starting from the rest. Thus, the values of the angles depend strongly on the granular material. The three angle values were adjusted within a range of feasible values according to the references ((), (), and ()):*
$$\delta_1 \in [1°, 22°], \quad \delta_2 \in [11°, 34°], \quad \delta_3 \in [3°, 23°].$$

*In the present paper we have employed the values*

$$\delta_1 = 6°, \quad \delta_2 \in [17°, 30°], \quad \delta_3 = 12°,$$

*for the three benchmark problems which is consistent with the references. As noted in (), in general for real problems involving complex rheologies, smaller values of these parameters $\delta_i$ should be employed.*

*With regard to the sensitivity of the model to parameter variation, an appropriate sensitivity analysis can be performed, as it is done in (). However, the aim of the present work was to prove if the non-hydrostatic model couple with the granular model was able to accurately reproduce the three benchmarks considered. A set*

[Figure]

*of tools can be developed to study the sensitivity to varying input parameters by adapting the ideas in (), but this is not the purpose of this paper.*

The parameter denoting the buoyancy effect is just a tuning parameter if the author cannot link it to the status of real landslide.

*We agree. For field case problems, $r = 0.5$ is usually taken, and then the parameter is tuned based on available field data. In general, the complexity of the rheology introduces a difficulty that is always present on the modelling as well as on the tuning of the parameters. Moreover, the more sophisticated is the model (considering, for example, the rheology of the material), the more input data -that may be unknown- will be required.*

What is the model user to set up this parameter If the user does not know any field or experimental observations ahead of time?

*Although the final user has to adjust some model input parameters, within a range of acceptable value (such as the aforementioned angle frictions), we remark the simplicity of the final numerical model proposed, and thus, on the efficient of its GPU implementation. This allows performing uncertainty quantification (see ()) on a few parameters, and investigating the sensitivity to them varying on small ranges (as in ()).*

*When field or experimental observations are available, a different approach is proposed in () where an automatic data assimilation strategy for a similar landslide non-hydrostatic model is proposed. That strategy can be also adapted for the present model.*

*Finally, we would like to emphasize on the simplicity of the granular model, the extremely overall efficient implementation, and the good results obtained for representing both water and granular motions. The very good computational performance achieved by the proposed numerical model allows the possibility of considering, for future works, data assimilation, uncertainty quantification or*

*performing sensitivity analysis on the parameters.*

*At the beginning of section 5, just before presenting benchmark results, a paragraph has been added, containing the answer to the second main point raised by the reviewer and describe here in our answer.*

Minor suggestions:

1. I am not sure if the authors use of the term "two-phase model" is correct or not. Most of the time only the model solving both the fluid phase and solid phase of the landslide is called "two-phase model".

   *In this paper we have two phases: water and granular material. Therefore, we consider two phases in the sense that two different fluids are modeled. However, in fact there is no mixing between them.*

   *We have found in the bibliography this terminology used exactly in the same way. For example, just to mention one: in two initial sentences of the Introduction of Drew (1983) "Mathematical model of two-phase flow" in Ann. Rev. Fluid Mech.(\*), you can read "Dispersed two-phase flows occur in many natural and technological situations. For example, dust in air and **sediment in water**...".*

   (\*)`https://www.annualreviews.org/doi/abs/10.1146/annurev.fl.15.010183.001401`

   *After many readings of reviewer's comment, and not really understanding his/her point, and after thinking about it, we think that the reviewer is meaning that "**Most of the time** only the model solving **OR** the fluid phase **OR** the solid phase of the landslide is called "two-phase model". (with BOTH in the sentence we do not really understand the question). We have looked for all the appearances of the term "two-phase model" in the paper and it appears four times. Two of them (at*

*the beginning of section 3 and in the conclusions) where, to our understanding, they perfectly fit the meaning of a two-phase model. The other two times, is true that the term is referring only to one component of the coupled system. In these two appearances, the term two-phase (in lines L195 and L219 of our current version) has been removed.*

2. In Line 54 the authors mention particular difficulty of the BP6 simulations. If they do not say somewhere what the difficulty is, they can simply delete the words in the brackets.

   *Done. We have delete the words in the brackets, as suggested.*

3. Both BP5 and BP6 provide the slide shape at different time in the experiments. I would recommend the authors to compare the slide shape of simulation results to the observations in Figure 4 and Figure 8.

   *In () some snapshots of the landslide evolution are shown at different time-steps that can be compared with Figure 4 corresponding to benchmark 4. It can be seen that the location of the landslide front is well-captured, but there is some mismatch of the landslide shape at the front, mainly due to the simplicity of the landslide model considered here. In particular, we consider that the density remains constant in the landslide layer during the simulation, this is not true due to the water entrainment into the slide material.*

   *A similar comparison can be made between Figure 8 (BP5) and some snapshots of the landslide evolution that can be found in () for the benchmark problem 5.*

   *As far as we know, no data is publicly available on the evolution of the slides, except for the aforementioned snapshots that can be found in (; ) or in the website (). Using these figures would require asking for permission to use them. Instead, we have included some comments about this in the final version of the paper for the interested reader that could check the cited references.*

4. The symbol $r$ is used twice.

   *It is true, but extremely difficult to avoid such coincidence as both symbols are really standard notations. One denotes the slide liquefaction and the other the wave gauge locations. We have changed this second $r$ to $R$ for Radius (hopefully not appearing anywhere else in the present work).*

5. Some typos: 2020a in line 17; duplicate sentences from line 276 to 280.

   *Both typos have been corrected.*

**References**

M. Brunet, L. Moretti, A. Le Friant, A. Mangeney, E.D. Fernández-Nieto, and F. Bouchut. Numerical simulation of the 30–45Âăka debris avalanche flow of Montagne Pelée volcano, Martinique: from volcano flank collapse to submarine emplacement. *Natural Hazards*, 87(2):1189–1222, 2017.

A.M. Ferreiro-Ferreiro, J.A. García-Rodríguez, J.G. López-Salas, C. Escalante, and M.J. Castro. Global optimization for data assimilation in landslide tsunami models. *Journal of Computational Physics*, 403:109069, 2020.

J. Garres-Díaz, E. D. Fernández-Nieto, A. Mangeney, and T. Morales de Luna. A weakly non-hydrostatic shallow model for dry granular flows, 2020.

J.M. González-Vida, J. Macías, M.J. Castro, C. Sánchez-Linares, S. Ortega, and D. Arcas. The Lituya Bay landslide-generated mega-tsunami. Numerical simulation and sensitivity analysis. *Nat. Hazards Earth Syst. Sci.*, 19:369–388, 2019.

S.T. Grilli, M. Shelby, O. Kimmoun, G. Dupont, D. Nicolsky, G. Ma, J.T. Kirby, and F. Shi. Modeling coastal tsunami hazard from submarine mass failures: effect of slide rheology, experimental validation, and case studies off the US East Coast. *Natural Hazards*, 86(1):353–391, 2017.

LTMBW. Landslide Tsunami Model Benchmarking Workshop, Galveston, Texas, 2017. http://www1.udel.edu/kirby/landslide/index.html, 2017. Accessed: 2020-12-08.

A. Mangeney, F. Bouchut, N. Thomas, J. P. Vilotte, and M. O. Bristeau. Numerical modeling of

self-channeling granular flows and of their levee-channel deposits. *Journal of Geophysical Research: Earth Surface*, 112(F2), 2007.

O. Pouliquen and Y. Forterre. Friction law for dense granular flows: application to the motion of a mass down a rough inclined plane. *Journal of Fluid Mechanics*, 453:133–151, 2002.

C. Sánchez-Linares, M. de la Asunción, M. Castro, J.M. González Vida, J. Macías, and S. Mishra. Uncertainty quantification in tsunami modeling using multi-level monte carlo finite volume method. *Journal of Mathematics in Industry*, 6, 12 2016.

S. Viroulet, A. Sauret, and O. Kimmoun. Tsunami generated by a granular collapse down a rough inclined plane. *EPL (Europhysics Letters)*, 105(3):34004, 2014.

---

## Author Comment (AC2) · 9 Dec 2020

**General Comments:**

As was the case for the first part of this paper, this is once again an informative paper that allows the modeling community to get a feeling of the reliability of the simulations performed with the Landslide-HySEA and Multilayer-HySEA models when used for real-life hazard assessment studies. This work presents results of modeling 3 of 7 benchmark problems proposed by the National Tsunami Hazard Mitigation Program (NTHMP). The three problems are based on data collected via experimental studies on tsunami generation by 2-D and 3-D, deformable slides from aerial and subaerial initial positions. The Landslide-HySEA version of the code is used to represent the slide dynamics by means of a Savage-Hutter approach. This is coupled with the Multilayer- HySEA version to capture the dispersive dynamics of the hydrodynamic phase.

*To be more precise both models (the Landslide-HySEA and the Multilayer-HySEA models) are independent models and they are not coupled one with the other. They share the same model (when granular slides are considered): a Savage-Hutter model, but both models are implemented independently and provide a different modelization of the fluid, while for the slide material they share the same Savage-Hutter model. For the fluid layer the Landslide-HySEA model implements a one-layer SW equations system while the Multilayer-HySEA model considers a multi-layer approach. In both cases with or without dispersion for the fluid. In the current work, we benchmark the dispersive Multilayer-HySEA model. The Landslide-HySEA model is described here in order to provide an introduction from a simpler model, not able to reproduce the experiments proposed, and as motivation for the need to use a more complex model able to produce realistic simulations. A sentence has been added in the Abstract to clarify this and explaining the reason why we introduce first the Landslide-HySEA model.*

The topic and results presented in the paper are within the scope NHESSD topics. The paper provides a sufficiently (perhaps too much) detailed description of both, the governing equations modeled in the code and the numerical algorithm implemented to resolve the system. Additional references are provided for readers interested in additional details. Given the complexity of the mathematical modeling and numerical scheme employed to solve the equations, one wonders whether the reader should be referred to a separate publication for that explanation, and the manuscript could be focused on the modeling setup and results alone. The authors provide an adequate literature review of pre-existing validation efforts in the introductory section of the report. They also provide a description of the numerical implementation of the laboratory experiments used for the validation. The results of all three experiments are presented in a clear and concise manner. I have not really found any major issues with the paper and I am ready to recommend publication with very minor modifications:

**Specific Modifications:**

- pp1 l.7: Here and in another location(s) in the paper the term "approved" is used to refer to the NTHMP process of testing codes. It should be corrected to "validated" or "tested" as the NTHMP does not "approve" or "certify" any models. Please check with NTHMP for clarification if needed.

  *The authors have checked this and found out that the NTHMP thoroughly used the term "approved" in the past but, it is true, that now this term does not appear in its web page anymore. Instead they moved to a lighter qualification of "Models that meet defined criteria for NTHMP Modeling and Mapping" and simply "Benchmarked Tsunami Models" (following NTHMP standards). Nevertheless, there exists a document of July 2015 entitled: "The NTHMP Tsunami Inundation Model Approval Process", describing all the steps in the process of "Approval" by the NTHMP. Finally, as far as authors know the term "approved" was changed to "accepted" in February 2016, in the document "The NTHMP Tsunami Inundation Model Benchmarking and Acceptance Process" in:*
  *https://nws.weather.gov/nthmp/documents/NTHMPTsunamiInundationModelAcceptance.pdf*
  *In any case, we agree with the referee, and we will no longer use the term "approved" and will refer as "accepted", "benchmarked" or "validated following the NTHMP standards".*

- pp1 l15-17: The authors mention the workshop consisted of 7 benchmark problems (3 were presented in Part I of the paper and 3 in this Part II), it would be good to explain if the missing problem was attempted and what results were obtained.

  *The seventh benchmark problem is the field case in Port Valdez 1964. It is the benchmark that we work the most for the Workshop and that we finished and complete for January 2017, but we did not take the time to write the corresponding paper. Now we have nearly finished the document and we expect to submit pretty soon. Once finished this revision...*
  *We briefly mention this and give a reference to our contribution to the NTHMP report, where the results corresponding to this seventh benchmark are collected.*

- pp 8, l181-182: Please, specify what boundary condition is applied where for each of the three BCs specified in the equations.

  *Done. We have specified in the paper the corresponding boundary conditions that were applied for each of the three benchmark problems.*

- pp14, 18, 23: For all benchmark problems, please specify how parameters (r, na, nm, ...) were selected or whether they were provided with the data. Also explain how Dx (delta x), Dy (delta y) is selected

  *The parameters involved at each simulation are:*

  $$g, \ r, \ n_a, \ n_m, \ d_s, \ \delta_i, \ \beta, \ and \ \gamma.$$

  *The parameters $g$, $r$, $n_m$, and $d_s$ are related to physical settings given at each experiment. $\beta$ and $\gamma$ are empirical parameters that were chosen as in the seminal paper of [11].*

  *The friction angles $\delta_1$, $\delta_2$ are characteristic angles of the material, and $\delta_3$ is related to the behavior of the slide motion when starting from the rest. Thus, the values for these parameters strongly depend on the granular material. The values for these three parameters were adjusted within a range of values found in references as [1], [9], or [11]:*

  $$\delta_1 \in [1°, 22°], \quad \delta_2 \in [11°, 34°], \quad \delta_3 \in [3°, 23°].$$

*In the present paper we have employed the values*

$$\delta_1 = 6°, \quad \delta_2 \in [17°, 30°], \quad \delta_3 = 12°,$$

*for the three benchmark problems, a choice that is consistent with the values provided in the literature. As noted in [9], in general for real problems involving complex rheologies, smaller values of $\delta_i$ should be employed.*

*We have added a paragraph in Section 5 to explain all this in detail.*

*Concerning how $\Delta x$ and $\Delta y$ were selected, here $\Delta x = \Delta y$ for all benchmark problems, and thus introducing the same numerical diffusion in both directions. However, in the case of pure one-dimensional domains problems, as BP4 or BP5, the effect of the y-resolution does not matter, and we just set the number of cells on the y-direction to be simply one.*

*The length of $\Delta x = \Delta y$ is consistent with the number of cells per wave used in other non-hydrostatic works that include dispersive effects (see, for instance, [2], [3], [5], [6], [7] and references therein). For instance, the reviewer is referred to [7] were authors employ the same grid resolution as here for BP6.*

*Moreover, we have performed a sensitivity analysis for the grid resolution, and we have observed that results presented in this work were well converged when using different decreasing size resolution.*

*In brief, the values of $\Delta x = \Delta y$ employed here ensures well-converged numerical results, and that waves can be well-represented maintaining a good balance between numerical results and computational efficiency.*

*Concerning the number of layers, $n_a$, and according to our experience, we could say that the one-layer model may produce wrong amplitudes and frequency waves, at least in areas close to the landslide generation (see for example the next paragraph in this answer). Model results can be improved by adding more layers. Typically, simulations with three or four layers will be accurate enough in most situations. That could be well explained according to the multi-layer model's dispersion relation (see [4]). See also [8] where numerical experiments with rigid landslides evidence that, in some situations, a simple one-layer system may produce wrong results in both amplitude and frequency dispersion. That is even more evident when steep gradients on the topography are involved and that the model must be accurate enough for the linear shoaling dispersion relation.*

*We have include the required information on the time steps and also number of layer in the new text.*

- pp17 l313: Please, correct units of density $(km/m^3)$

  *This was already corrected in the current version of the manuscript uploaded in the system*

- pp17, l316: Please, replace "consists in" with "consists of", wherever it appears in the paper.

  *Four matches were found and changed.*

- pp23, l370: It would be interesting to know whether the non-dispersive case of 1-layer was attempted and how the results would compare with the multi-layer cases. If available, please add.

*We have those for the benchmark problem 6, the results obtained using the one-layer non-dispersive model are still good and similar to the two-layer model, but for the run-up (see here, Figures 2, 3 and 4). Nevertheless it can be observed a frequency mismatch on the time series, that it is not the case if two or more layers are used. We have not included this comparison in the paper. If the reviewer considers that this is interesting, we could add these figures in the paper.*

*Moreover, we have performed benchmark problem 4 with the numerical model proposed by setting the number of layers equal to one. The comparison with the experimental data can be seen in this answer in Figure 1. It can be seen the classic behavior of a dispersive system that does not have an accurate enough dispersion relation for the water waves involved here. It can be seen how the amplitude, as well as the frequency wave train, are not well represented. A similar comparison for a rigid landslide problem is shown in [8].*

- p23, l371-376: The description of how the slide is initiated is unclear. Please, explain with more detail. Is the entry velocity specified? If not, how is it reached? What is the function of the pneumatic pistons?

  *The entry velocity was already specified, and we have better detailed in the paper:*
  *"In [10], it is shown that for this test case, the landslide box reached a velocity of $v_b = 2.3 \cdot \sqrt{g \cdot 0.6} = 5.58$ m/s. Thus, the initial condition for the water velocities is set to zero:*

  $$u_i = 0, \ i = 1, 2, \ldots, L$$

  *and for the landslide velocity is set to the above-mentioned constant value:*

  $$u_s = 5.58, \text{wherever } z_s > 0,$$

  *for the x-component. The y-component of the landslide velocity was initially set to zero."*

- pp24, Figure 10: I would suggest using more distinguishable colors for the lines in the top panel, it is hard to tell the Grain from the Grain Velocity lines.

  *Done in the next version of the manuscript.*

  -Please, be more detailed in the legend, specify what magnitude is represented by "Grain".
  *We have changed "Grain" to "Granular slide". Thus, "Granular slide", "Free Surface" and "Bottom", refer to the location of these surfaces and, in particular, the "Granular slide" legend refers to the location and geometry of the simulated granular slide. Figure captions have also being changed accordingly. "Grain velocity" has been changed to "slide velocity".*

  -Does the vertical axes represent position or velocity? Perhaps, the left axes should be used for distance and the right one for velocity?

  *Done.*

- p27, Figure 12: Please specify if number in top left corner refer to x-, y = (positions)

  *Done. This has been specified in the captions of the Figure 11 $(R, \theta°)$ and 12 $(x, y)$.*

Some stylistic corrections (these are some of the corrections needed, but not all, please scan the document for additional typos):

- pp4, l 99: Please correct to "initiative which the present work is based on"

  *Done.*

[Figure]

Figure 1: Numerical time series for the simulated water surface with one layer (in blue) compared with lab measure data (red) at wave gauges (A) WG1, (B) WG2, (C) WG3, and (D) WG4.

- pp6, l144-147: move: "..., the ratio r is also constant (rho_f and rho_s are also constants)" from line 147 to line 145.

  *We change the overall writing of a coupled of sentences here, now the text is: "where $\rho_s$ stands for the typical density of the granular material, $\rho_f$ is the density of the fluid ($\rho_s > \rho_f$) both constant, and $\varphi$ represents the porosity ($0 \leq \varphi < 1$). In the current work, the porosity, $\varphi$, is supposed to be constant in space and time and, therefore, the ratio r is also constant."*

[Figure]

Figure 2: BP6. One-layer. Cross-Section.

[Figure]

Figure 3: BP6. One-layer. Temporal series.

[Figure]

Figure 4: BP6. One-layer. Run-up.

- pp6, l154: Please replace "vertical variable" with "vertical coordinate".

  *Done.*

- pp9, l197: Correct: "The Savage-Hutter model that is used and...."

  *We do not see were is the correction. We slightly modify the wording and now it is written as: "The 1D Savage-Hutter model used and implemented in the present work is given by the system".*

- pp12, l238: Please, spell out "HLL"

  *Harten-Lax-van Leer. Done*

- pp13, l276-280: The first sentence is repeated almost literally. Please, correct.

  *Fully rewritten.*

- pp14, l304: What is meant by "no longer"?, Please word correctly.

  *I cannot find any "no longer" in the current version of the manuscript.*

- pp25, l405: Correct: "In can be..."

  *Done.*

- pp.28: l415: Please, correct: to "The present work aims at benchmarking the model..."

  *Done.*

- pp.28, l432: Correct to: "Savage-Hutter used here".

  *Done.*

- pp.29, l452: Correct to: "data compared with,.."

  *This part of the text was rewritten.*

**Bibliography**

[1] M. Brunet, L. Moretti, A. Le Friant, A. Mangeney, E.D. Fernández Nieto, and F. Bouchut. Numerical simulation of the 30–45 ka debris avalanche flow of Montagne Pelée volcano, Martinique: from volcano flank collapse to submarine emplacement. *Natural Hazards*, 87(2):1189–1222, Jun 2017.

[2] C. Escalante, T. [Morales de Luna], and M.J. Castro. Non-hydrostatic pressure shallow flows: Gpu implementation using finite volume and finite difference scheme. *Applied Mathematics and Computation*, 338:631 – 659, 2018.

[3] C. Escalante, E.D. Fernández-Nieto, T. Morales, and M.J. Castro. An efficient two–layer non–hydrostatic approach for dispersive water waves. *Journal of Scientific Computing*, 2018.

[4] E.D. Fernández-Nieto, M. Parisot, Y. Penel, and J. Sainte-Marie. A hierarchy of dispersive layer-averaged approximations of Euler equations for free surface flows. *Communications in Mathematical Sciences*, 16(5):1169–1202, 2018.

[5] James T. Kirby, Fengyan Shi, Dmitry Nicolsky, and Shubhra Misra. The 27 April 1975 Kitimat, British Columbia, submarine landslide tsunami: a comparison of modeling approaches. *Landslides*, 13(6):1421–1434, Dec 2016.

[6] J.T. Kirby, S.T. Grilli, C. Zhang, J. Horrillo, D. Nicolsky, and P. L.-F. Liu. The NTHMP landslide tsunami benchmark workshop, Galveston, January 9-11, 2017. Technical report, Research Report CACR-18-01, 2018.

[7] Gangfeng Ma, James T. Kirby, Tian-Jian Hsu, and Fengyan Shi. A two-layer granular landslide model for tsunami wave generation: Theory and computation. *Ocean Modelling*, 93(C):40–55, 2015.

[8] J. Macías, C. Escalante, and M.J. Castro. Multilayer-HySEA model validation for landslide generated tsunamis. Part I Rigid slides. *Submitted to Nat. Hazards Earth Syst. Sci.*, 2020.

[9] A. Mangeney, F. Bouchut, N. Thomas, J. P. Vilotte, and M. O. Bristeau. Numerical modeling of self-channeling granular flows and of their levee-channel deposits. *Journal of Geophysical Research: Earth Surface*, 112(F2), 2007.

[10] F. Mohammed. *Physical modeling of tsunamis generated by three-dimensional deformable granular landslides*. PhD thesis, Georgia Institute of Technology, 2010.

[11] O. Pouliquen and Y. Forterre. Friction law for dense granular flows: application to the motion of a mass down a rough inclined plane. *Journal of Fluid Mechanics*, 453:133–151, 2002.